# Finite Element Method and Cut Bar Method-Based Comparison Under 150°, 175° and 310 °C for an Aluminium Bar

José Eli Eduardo Gonzalez Duran [1,†] , Oscar J. González-Rodríguez [2,†],
Marco Antonio Zamora-Antuñano [3,†] , Juvenal Rodríguez-Reséndiz [4,*,†] ,
Néstor Méndez-Lozano [3] , Domingo José Gómez Meléndez [4,‡] and Raul García García [5]

1   Instituto Tecnológico Superior del Sur de Guanajuato, Guanajuato 38980, Mexico; je.gonzalez@itsur.edu.mx
2   Centro Nacional de Metrología (CENAM), Querétaro 76246, Mexico; ogonzale@cenam.mx
3   Departamento de Ingenieria, Universidad del Valle de Mexico, Querétaro 76230, Mexico;
    murck22@gmail.com (M.A.Z.-A.); nestor.mendez@uvmnet.edu (N.M.-L.)
4   Facultad de Ingenieria, Universidad Autónoma de Querétaro, Querétaro 76010, Mexico;
    juvenal@uaq.edu.mx
5   Departamento de Ingenieria, Universidad Tecnológica de San Juan del Río, San Juan del Río 76800,
    Querétaro, Mexico; rgarciag@utsjr.edu.mx
*   Correspondence: juvenal@uaq.edu.mx; Tel.: +52-442-192-12-00
†   These authors contributed equally to this work.
‡   This author have passed away.

**Abstract:** Analyses were developed using a finite element method of the experimental measurement system for thermal conductivity of solid materials, used by the Centro Nacional de Metrología (CENAM), which operates under a condition of permanent heat flow. The CENAM implemented a thermal conductivity measurement system for solid materials limited in its operating intervals to measurements of maximum 300 °C for solid conductive materials. However, the development of new materials should be characterised and studied to know their thermophysical properties and ensure their applications to any temperature conditions. These task demand improvements in the measurement system, which are proposed in the present work. Improvements are sought to achieve high-temperature measurements in metallic materials and conductive solids, and this system may also cover not only metallic materials. Simulations were performed to compare the distribution of temperatures developed in the measurement system as well as the radial heat leaks, which affect the measurement parameters for an aluminium bar, and uses copper bars as reference material. The simulations were made for measurements of an aluminium bar at a temperature of 150 °C, in the plane and 3D, another at 175 °C and one more known maximum temperature reached by a sample of the aluminium bar with a new heater acquired at 310 °C.

**Keywords:** cut bar method; thermal conductivity; finite element method; steady-state; heat lakes

---

## 1. Introduction

Thermal conductivity is a physical property of materials that measures heat conduction capacity. In other words, thermal conductivity is also the ability of a substance to transfer the kinetic energy of its molecules to adjacent ones or to substances with which it is in contact. In the International System of Units, the thermal conductivity is measured in W/(m K) equivalent to J/(m s K). There are several methods to measure the thermal conductivity of materials: the most conventional method for measuring thermal conductivity consists of two concentric metal spheres, of very small thickness

to minimise the heat capacity of the system. It has not been used with measurements greater than 300 °C [1]. The parametric study consists in obtaining the temperature distribution in the most insulating composite bar system for different operating conditions. As operating conditions, it refers to the temperature difference at the ends of the system, characteristics of the reference material, aspects of the insulating material and features of the sample materials [1,2]. The determination of the thermophysical properties of materials is essential in all processes where energy exchanges occur, in particular, heat. For the design, operation and maintenance of systems and equipment where the temperature is present, it is essential to know the value of these properties in particular of thermal conductivity. This property has an important effect on solid thermal conductive materials such as aluminium, iron, copper, its alloys, and new materials that are used to build equipment and machinery parts, such as automotive vehicle engines [1]. Thermal conductivity is also an issue related to the second law of thermodynamics or the law of entropy, which governs most of the phenomena that occur in the universe, by which it is estimated that any process that involves work increases the entropy of the universe (increases the disorder and chaotic movement of atoms and the temperature of existing molecules and grains). Thermal energy always flows spontaneously from highest to lowest concentration, or from hot to cold. This implies that heat transfer by conduction occurs from one body to another at a lower temperature or between areas of the same material but with a different temperature. Heat transmission involves an internal energy exchange, which combines potential energy and kinetic energy of electrons, atoms, and molecules: the higher thermal conductivity, the better the heat conduction. The inverse property is the thermal resistivity, which indicates that, at lower thermal conductivity, more heat insulation (more resistivity). Concerning potential energy, we can say that it is the mechanical energy associated with the location of a body in a field of forces or the presence of an area of effects within the body itself. The potential energy is the result that the system of forces that affects a given body is conservative, then, the total work on a particle is zero. The kinetic energy of a body, meanwhile, is what it has thanks to its movement. It is the work needed to achieve its acceleration from rest to a given speed. When the body reaches this energy throughout the acceleration, it maintains it unless it alters its speed. To return to the resting state, it is necessary to perform a dangerous job with the same magnitude. By heating matter, the average kinetic energy of its molecules increases, and this increases its level of agitation. At the molecular level, heat conduction occurs because the molecules interact with each other by exchanging kinetic energy without making global movements of matter. It should be mentioned that at the macroscopic level, it is possible to model this phenomenon by means of Fourier's law.

The CENAM implemented a system to measure thermal conductivity of solid conductive materials, and the design criteria were developed for the construction of the measurement system, which operates under the condition of heat flow in a permanent state. The system uses a reference material, which limits the accuracy of the method. An analysis of the system is carried out considering that there is axial and radial heat flow. In addition, the solid bar of material that can be evaluated, it has a hollow bar of insulating material. The problem to be solved is a bar composed of a reference conductor material at the longitudinal ends, and a test material depicted in the centre, the entire bar an element by an insulating material, it is considered that it is axial and radial flow and the physical dimensions of the problem are shown in Figures 1a, 2 and 3. Using the apparatus developed in the CENAM, two concentric cylinders are used, housing the material to be tested between them. Inside the smaller diameter cylinder is placed the heating resistance, which is covered with another cylinder to standardise the surface temperature. The temperature measurement is carried out on the outer and inner cylinders, using thermocouples for this. The method is used to measure thermal conductivity in materials solid conductors. To meet this need for measurement, the CENAM developed a system for measuring thermal conductivity in thermally conductive solid materials employing a secondary method. This work presents a comparison of certain experimental results using the cut bar and the finite element method (FEM), to obtain information that serves in the development of the new cut bar system, to extend its operating range up to 600 °C, under optimal operating conditions [2].

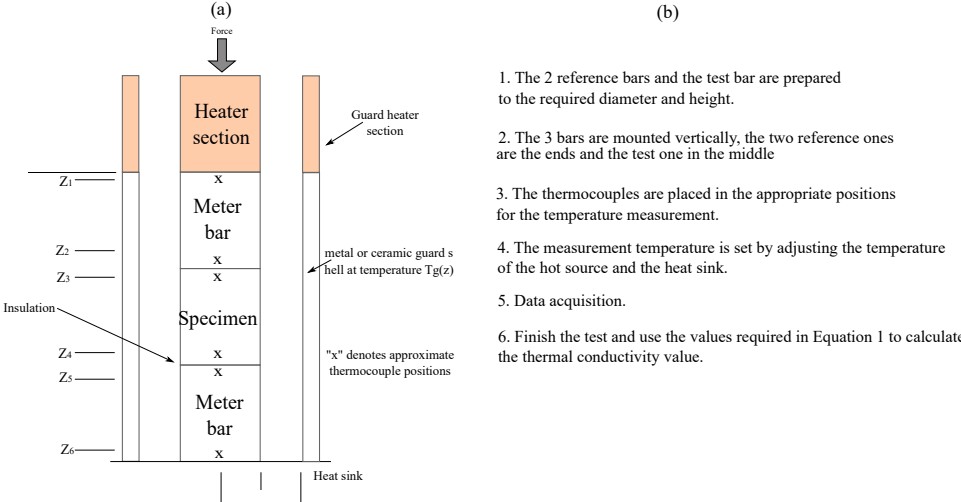

**Figure 1.** (**a**) Schematic of a Comparative-Guarded-Longitudinal Heat Flow System, indicating possible locations of temperature sensors (**b**) methodology for the experiment used in this work [1].

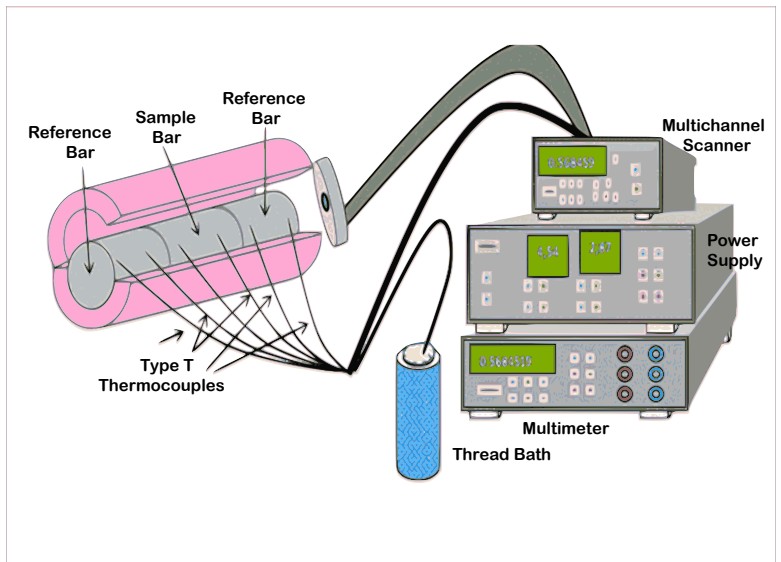

**Figure 2.** Cut bar method system [2].

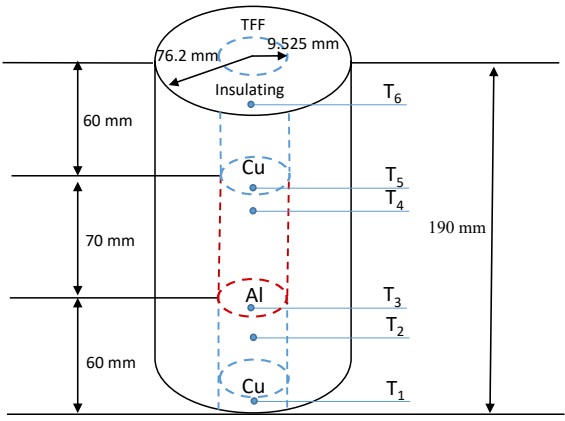

**Figure 3.** Diagram of the experimental model used.

## 2. Technique Background

The method determines the thermal conductivity of a sample using a reference material by a permanent state technique known as the concentric bar cut method [3]. The system consists of a bar with well-known properties, called reference bar, another bar with conductivity to be determined, called sample o test bar, and another reference bar. The composite metal bar is covered with an insulating material to prevent heat flow in the radial direction. At one end of the composite bar, a heat source is placed, and at the opposite end, there is a heat sink or cold source [4,5]. Then, employing temperature and length measurements, the conductivity of the sample material can be determined. Figure 1 shows a diagram of the composite bar system.

The arrangement diagram of the bars for the method used in this work is shown in Figure 1a. The reference bars are located at the ends and the test bar in the center of both. At one end of the bar array, a heat sink or cold source is located; at the other end, a heater that allows generating a temperature gradient, necessary for the determination of the thermal conductivity value. Marked with $x$ in Figure 1a is where the thermocouples indicated at a certain height in mm and designated according to the letter $z$ are located; the thermocouples type T were a fine wire of 0.6 mm diameter from OMEGA brand, calibrated by CENAM. $r_A$ indicates the radius of the bars used for the test and $r_B$ the radius of the insulation used in the test. A force is applied axially to improve the contact between the bars axially.

A brief description of the process performed to carry out the test is named in Figure 1b. Before starting the test the bars to be used need to meet the necessary diameter and height, as well as some flatness on the flat faces of the bars, after these perforations are made on the cylindrical face a few millimeters deep to house the thermocouples later. After finishing the bars, they are placed one above the other in the order, as appear in Figure 1a. Then the thermocouples are placed in each one of the sweepers made in the bars, then it is surrounded with the insulating material, and the guard is added; the axial force is applied to improve the contact. The next step is to adjust the operating temperature of the hot and cold source according to the measurement temperature at which the test is required to reach the operating temperature was used as a power supply, which supplies the necessary voltage to an electrical heater until it reaches temperature operation, which is registered by the thermocouple located in the hot source. For example, if a test temperature of 100 °C is required, the average temperature of the hot and cold source must be sought to be 100 °C. For example, the temperature of the hot source at 150 °C and the cold source at 50 °C, so the average temperature is 100 °C. In this way, several combinations can be generated. Once the operating temperatures have been adjusted, data acquisition begins, by a program in LabView developed by CENAM, where the signal of thermocouples are read by a multimeter and send to a PC to register its values. The values will be adequate when a the steady-state regime has been reached, it is known, because charts of temperature from thermocouples do not change, reach a constant temperature throw experiment in time. With the acquired data, Equation (1) is used to calculate the thermal conductivity value.

From the work in [2] it was found that the thermal conductivity of the sample is given by

$$\lambda_M = \frac{Z_4 - Z_3}{T_4 - T_3} \left[ \frac{\lambda_{R_1}}{2} \left( \frac{T_2 - T_1}{Z_2 - Z_1} \right) + \frac{\lambda_{R_2}}{2} \left( \frac{T_6 - T_5}{Z_6 - Z_5} \right) \right] \tag{1}$$

where $\lambda_M$ is the thermal conductivity of the sample. $\lambda_{R1}$, and $\lambda_{R2}$ are the thermal conductivity of reference materials 1 and 2. $T_i$ is the temperature in each of the $Z_i$ positions where the thermocouples are placed. Subscripts 1 and 2 refer to the first reference bar, 3 and 4 to the sample under measurement and 5 and 6 to the second reference bar.

If the distances between the thermocouples of each bar are equal and the reference material is the same for the two bars, so from Equation (1) which the thermal conductivity of reference materials leave the equation as a common term. Taking into account that the distances are also equal, $(Z_2 - Z_1)$ is the same that $(Z_6 - Z_5)$ then leave the parenthesis so with $(Z_4 - Z_3)$ obtain unity. Then, to simplify

is rewritten $(T_4 - T_3)$ such as $\Delta T_2$, rewrite $(T_6 - T_5)$ such as $\Delta T_3$ and $(T_2 - T_1)$ such as $\Delta T_1$. Then, Equation (1) is reduced to

$$\lambda_M = \frac{\lambda_{R_2}}{2} \left( \frac{\Delta T_1 + \Delta T_3}{\Delta T_2} \right) \tag{2}$$

where $\lambda_{R_2}$ is the same the $\lambda_{R_1}$ because the reference material are equal. $\Delta T_1$ and $\Delta T_3$ are the difference among each reference bar and $\Delta T_2$ is the difference of temperature of the test bar. The cold source or heat sink is constituted by a 10 cm diameter copper plate that has a 10 mm diameter copper tube coil welded through which a fluid such as ethylene glycol flows from a bath of controlled temperature. One of the surfaces is in contact with one end of a reference bar and the other part in an insulated container [6]. The recirculation bath can maintain the temperature of the cold source between $-30\,^\circ$C and $60\,^\circ$C.

The reference material bars are 99.999% high purity copper with a diameter of 19.1 mm and a length of 60 mm. The composite bar is surrounded by a 100 mm diameter polyvinyl chloride (PVC) pipe, the inside of which contains 50.8 mm thick fiberglass.

The measurement system has seven calibrated type T thermocouples. The electromotive force (EFM) of each thermocouple is measured with a digital high-accuracy multimeter of $8\frac{1}{2}$ digits model 3458A from Agilent Technologies and aided by an 8-channel scanner keithley 7001 both manufactured in USA. CENAM developed a computer program for the control, reading, and recording of data. The program to acquire data works with a graphic interface developed in LabView, where is registered tension measurement of each thermocouple and through of coefficient obtained from calibration and with the Newton–Raphson method is converted tension measurement to a temperature value. The EFM was measurement by scanner and multimeter connected to a PC. That value is introduced in a subprogram, where is converted to each temperature value from each thermocouple used in the experiment. With the distance between thermocouples of each bar and the thermal conductivity from reference bar is calculated the thermal conductivity of the bar under test. In the front panel of Labview developed by CENAM, are showed temperature of each thermocouple used, temperature of hot and cold source, constants from calibration of thermocouples used, graphs of $\Delta T_1$, $\Delta T_3$ and $\Delta T_2$ as well as the thermal conductivity value. Figure 2 shows a schematic of the measurement system [7].

## 3. Methodology

The development of the experiment was carried out using the following parameters. Two copper bars of 60 mm in length were used, and one aluminium bar of 70 mm length. The three bars have a diameter of 19.05 mm.

As an insulator to reduce radial heat leaks, glass fiber with thermal conductivity of 0.046 W/mK was used. The thermocouples were placed in such a way that there was a distance of 40 mm for the copper bars and 50 mm for the aluminium bar [8]. These were placed on the outside of the bars, on the surface, based on the work in [9], the standard test method for thermal conductivity of solids by means of the guarded-comparative-longitudinal heat flow technique. The configuration of the bars is depicted in Figure 3.

*Computational Model Setup*

It was made a model by aided computer design under dimensions illustrated in Figure 3, and due to its symmetry, a 2D model was done and another with azimuthal symmetry [10]. The temperature of the hot source (HST) was extracted from the data of experimental results; the same for the cold source temperature (CST) which were introduced as boundary conditions in the finite element model, the properties of the material, in the case copper, being a reference material, Equation (3) was used

$$\lambda_{Cu} = 416.3 - 0.05904T + 7.087 \times 10^7 / T^3 \tag{3}$$

Equation (3) was obtained from [9] because in the standard are published thermal conductivity of some materials considered such as meter bar reference materials for the cut bar method.

Mesh for the Models Used

ANSYS 18 software with the Mechanical APDL (Parametric Design Language) user interface and thermal module for the simulations of this work was chosen. For the flat model, it was used a PLANE 77 element of eight nodes, and it has one degree of freedom, temperature, at each node, and applies to a 2-D, steady-state or transient thermal analysis and a SOLID 90 element of 20 nodes with a single degree of freedom, temperature, at each node for the 3D model. The mesh for 3D and 2D models are shown in Figure 4a,b, respectively [11,12].

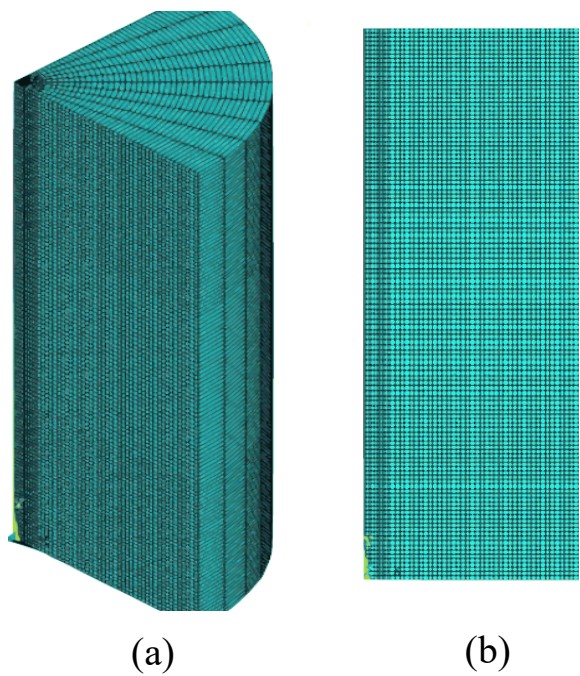

(a) (b)

**Figure 4.** Final mesh (**a**) for the 3D model and (**b**) the 2D model used in this work.

The mesh generated for the 3D model is depicted in Figure 4a, where azimuthal symmetry was implemented. For the 2D model, the mesh that was generated is shown in Figure 4b, where symmetry was used on the $y$ axis. For loads, were used temperature values from the experimental results and heat flux 0 on the boundaries where the insulating material is presented. To reduce resources and computational time.

## 4. Results

The results for different temperatures of the aluminium sample selected in the experiment are described below, which were at 50 °C and one at 175 °C and the behaviour at an HST of 600 °C. It is because the heater that was acquired reaches a maximum operating temperature value of 600 °C. Only the points where the thermocouples are located were compared, both in copper and aluminium bars. The tables include material properties and initial conditions applied to the model of the finite element method. Also, this work shows a comparison of the results obtained in the simulation using ANSYS and the data acquired experimentally [13,14].

*4.1. 2d and 3d Analysis at a Temperature of* 150 °*C*

In Table 1 appears the boundary conditions applied for the 150 °C test. Also, in that table the material properties were introduced in the simulation the values, like 214.4 W/mK for aluminium,

386 W/mK for copper and 0.044 W/mK for fibreglass. The values for temperature loads used in the model in ANSYS were 279.5 °C for the hot source and 20 °C for the cold source.

**Table 1.** Boundary conditions applied for $T_{BAR}$ = 150 °C.

| Aluminum (Al) | → | 214.4 W/mK | $T_{FC}$ → 279.5 °C |
|---|---|---|---|
| Copper (Cu) | → | 386 W/mK | $T_{FF}$ → 20 °C |
| Fiberglass | → | 0.044 W/mK | |

Figure 5 shows the results obtained for the 150 °C test in the 2D model. In the distribution of temperature, it can be seen where the hot source, in red colour, and the cold source, in blue colour, are, in Figure 5a, the path begins with the origin and 289.6 °C on the coordinate axis, which corresponds to the point located in the hot source. The graphic represents the vertical line, where the model presents symmetry [15]. The graph in Figure 5b points out three slope changes, indicating the two types of materials since their thermal conductivities are different. Then the first slope from left to right is equal as the third slope since they are the same material and therefore have the same thermal conductivity value.

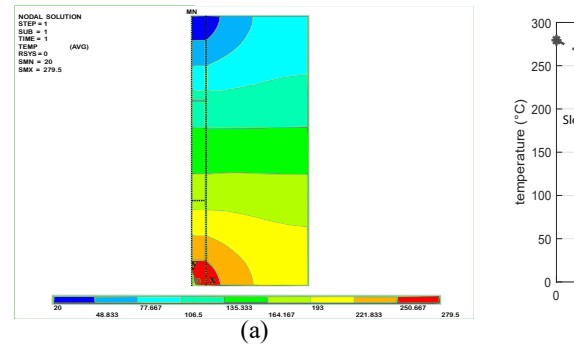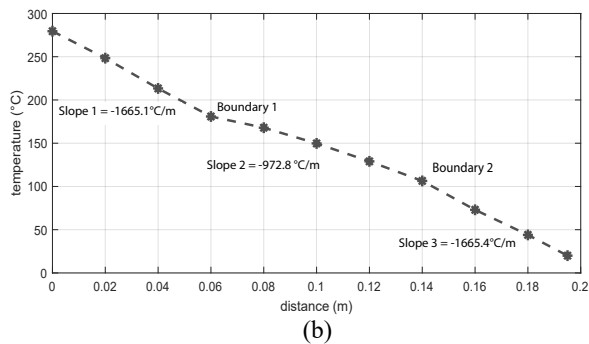

**Figure 5.** Results for 2D analysis at a test temperature of 150 °C. (**a**) temperature distribution in °C. (**b**) graphic of temperatures of the symmetry line.

The results of using a 3D model with azimuthal symmetry are illustrated in Figure 6; the differences are notorious compared with the temperature distribution image concerning the 2D model. In the graph, the changes are smoother; however, changes in the slopes are more defined. Also, we can see the point where contact exists in each metallic bar [16].

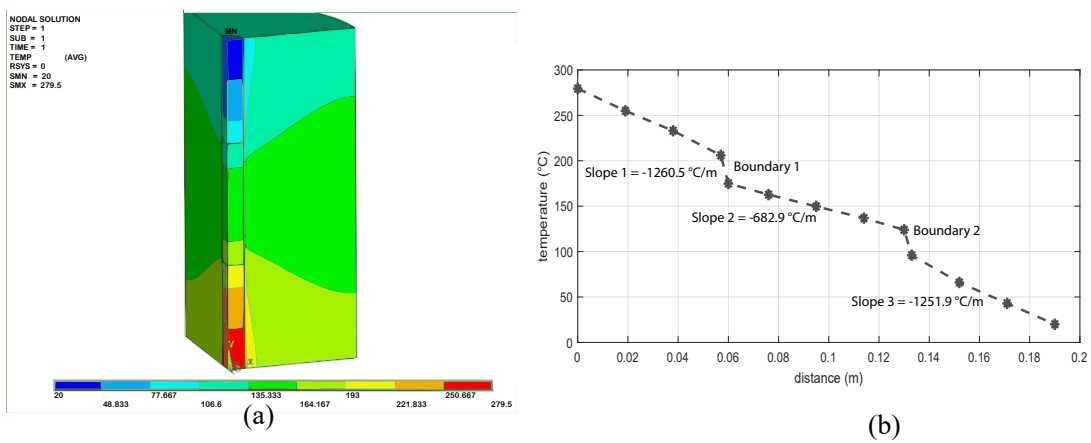

**Figure 6.** Results for 3D analysis at a test temperature of 150 °C. (**a**) Temperature distribution in °C. (**b**) Graphic of temperatures of the symmetry line.

In Figure 6b it can be observing a better definition of slope change between each bar. Even with that inflection point in the graphs, it can be obtained the temperature reached the junction of each bar. The transition between the boundary of each material is due to in analysis 2D, the interface is a line, but in 3D analysis, there are two surfaces, so in this case, ANSYS take in account radial heat transfer through surfaces.

However, Figure 7 shows the graph where the results are compared between 2D, 3D models, and the points that represent the thermocouples where experimentally are located in the metallic bars. The deviations are more significant near the borders, where the cold and hot sources are located. From the 2D and 3D simulations performed, the temperature values were extracted at the points where the thermocouples are experimentally located. The differences between these values were calculated to obtain the maximum and the minimum deviation between the results obtained from the simulation and the experiment. Therefore, the most substantial variance was around 20 °C, and the lowest was 1.6 °C.

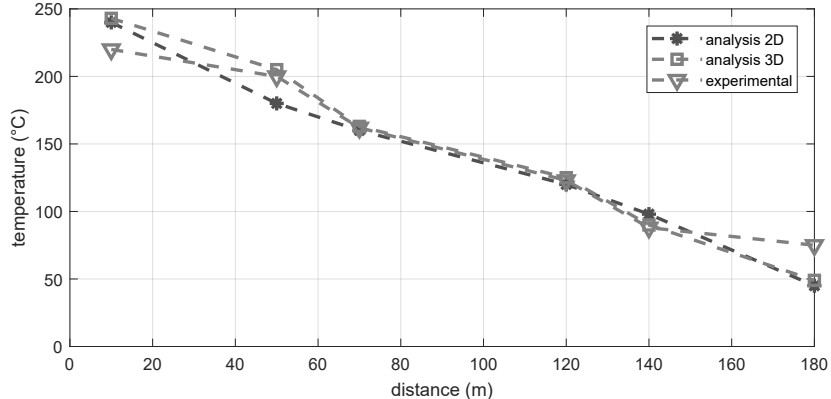

**Figure 7.** Comparison of results for 2D, 3D and experimental analysis at a test temperature of 150 °C in thermocouple positions.

### 4.2. 2d and 3d Analysis at a Temperature of 175 °C

The boundary conditions applied for the 175 °C test are shown in Table 2.

**Table 2.** Boundary conditions applied for $T_{BAR}$ = 175 °C.

| Aluminum (Al) | → | 214.4 W/mK | $T_{FC}$ → 339 °C |
|---|---|---|---|
| Copper (Cu) | → | 386 W/mK | $T_{FF}$ → 20 °C |
| Fiberglass | → | 0.044 W/mK | |

Figure 8 indicates the results obtained for the 175 °C test in the 2D model. In the temperature distribution, it is possible to observe where the hot source, in red, and the cold source, in blue, whose gradients are very similar to the test temperature at 150 °C. Figure 8b shows a graph where begins with the origin and 339 °C on the ordinates axis, which corresponds to the point where the hot source is, which represents a maximum temperature reached by the hot source. The graph describes the vertical line where the model presents symmetry. The graph represents the nodes that are on the vertical line of symmetry of the model. As in the previous case, it points out three slopes because there are two section changes, in this case, the slopes are of higher value because the operating temperature for the heater was higher than in the previous case.

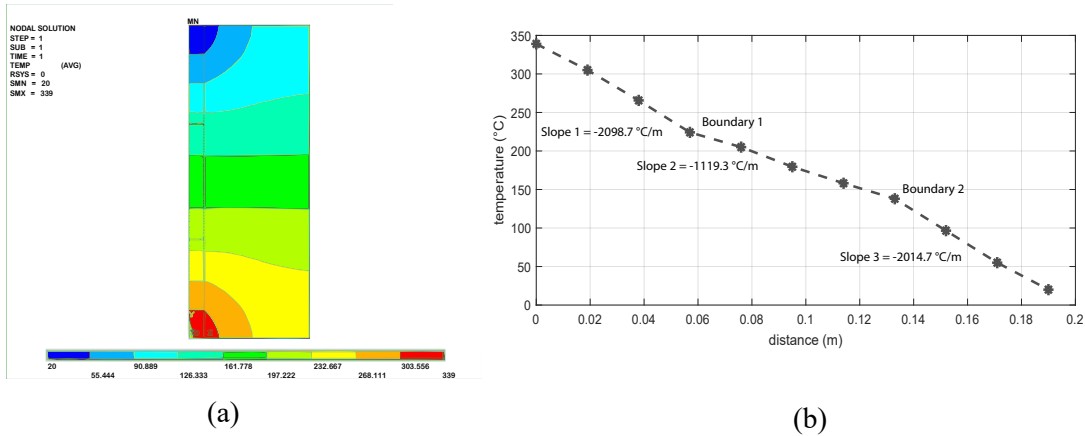

**Figure 8.** Results for 2D analysis at a test temperature of 175 °C. (**a**) temperature distribution in °C. (**b**) graphic of temperatures of the symmetry line.

Figure 9 indicates the results of using a 3D model with azimuthal symmetry. The differences are notoriously comparing the temperature distribution image concerning the 2D model. In the graph, the changes are smoother, showing the variation of the section between the copper reference bar and the aluminium test. The changes are due to the temperature gradient and the different values of the thermal conductivity of each bar [17,18]. In this case and the previous one, the deviations concerning the experimental results are more significant near the cold source, and, in both comparisons for the case of the hot source, the finite element method predicts and for the cold source sub predicts the actual values according to those obtained in the experiment, which means that heat leaks are present.

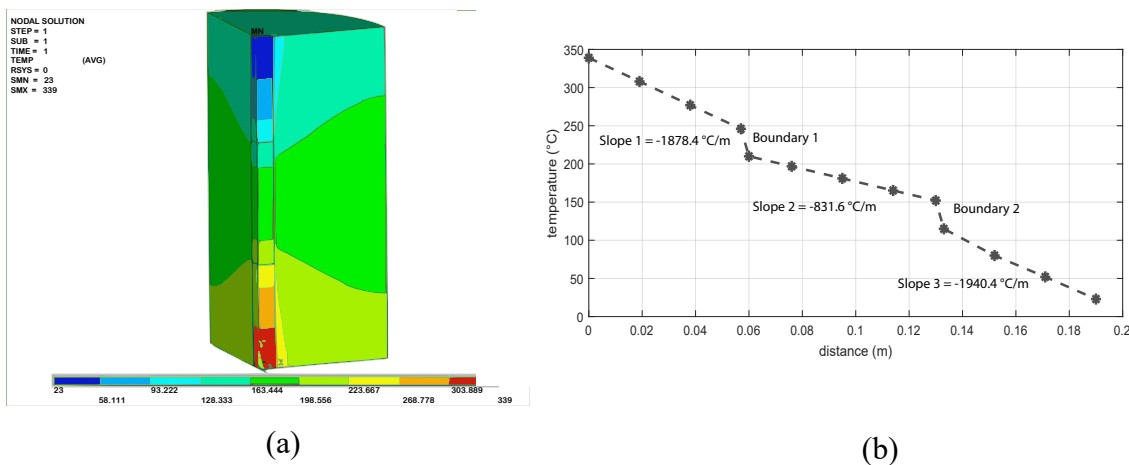

**Figure 9.** Results for 3D analysis at a test temperature of 175 °C. (**a**) Temperature distribution in °C. (**b**) Graphic of temperatures of the symmetry line.

However, Figure 10 shows a result comparison between 2D, 3D models, and the experiment. Where the deviations are more significant near the borders, where the cold and hot source are located. Because in that zones exists the most more significant gradients with the surroundings, because temperature laboratory is 22 °C. The most significant deviation for this case was around 37 °C and less than 7 °C.

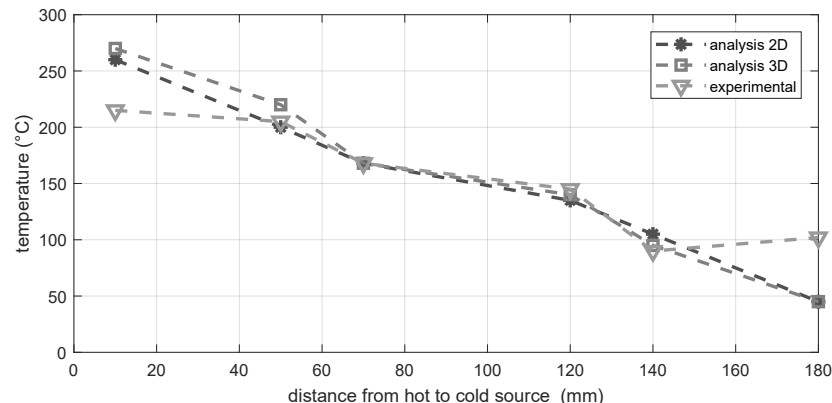

**Figure 10.** Comparison of results for 2D, 3D and experimental analysis at a test temperature of 175 °C in thermocouple positions.

### 4.3. 2D and 3D Analysis at a Temperature of 310 °C

Because a new heater was purchased, which operates at a maximum temperature of 600 °C, it is essential to know the temperature that the aluminium bar reaches, and by consequently evaluate if the conditions of the equipment are adequate to this new working temperature [19]. In Table 3 appears the boundary conditions applied for the 310 °C test, where it is observed that the maximum temperature reached by the new heater at 600 °C.

**Table 3.** Boundary conditions applied for $T_{BAR}$ = 310 °C.

| | | | |
|---|---|---|---|
| Aluminum (Al) | → | 200 W/mK | $T_{FC} \rightarrow$ 600 °C |
| Copper (Cu) | → | 365.74 W/mK | $T_{FF} \rightarrow$ 20 °C |
| Fiberglass | → | 0.044 W/mK | |

Figure 11 indicates the results obtained for the 310 °C test in the 2D model. In the distribution of temperature, it is possible to observe the hot source, in red, and the cold source, in blue, whose gradients are equal to the test temperature at 150 °C. However, the values of temperature in each zone are higher than the last case. In Figure 11b, the graph begins with the origin and 600 °C on the ordinate axis, which corresponds to the point located in the hot source as we can see slopes are greater than the last case because the temperature is higher. The graph represents the vertical line where the model presents symmetry.

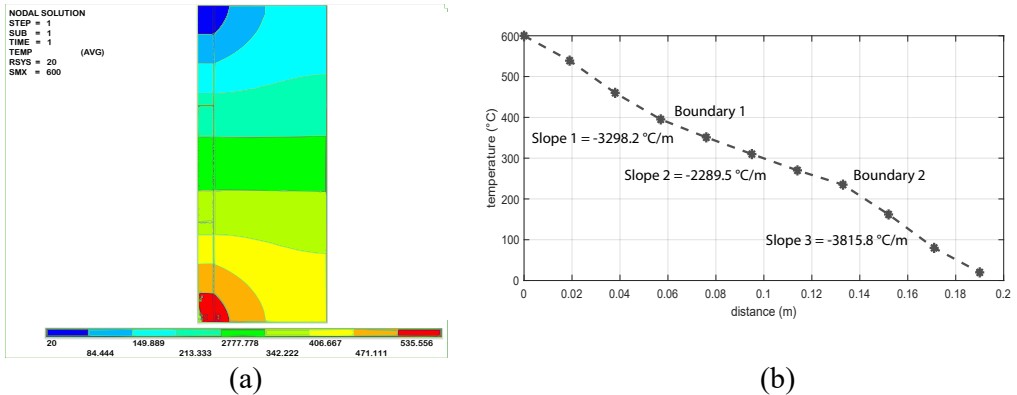

(a)  (b)

**Figure 11.** Results for 2D analysis at a test temperature of 310 °C. (**a**) Temperature distribution in °C. (**b**) Graphic of temperatures of the symmetry line.

Figure 12 indicates the results from the 3D model with azimuthal symmetry. The differences are notorious by comparing the temperature distribution image concerning the 2D model. The graph points out the temperature reached by the sample aluminium bar is 310 °C. Ideally, the temperature of the sample bar reached with the new heater.

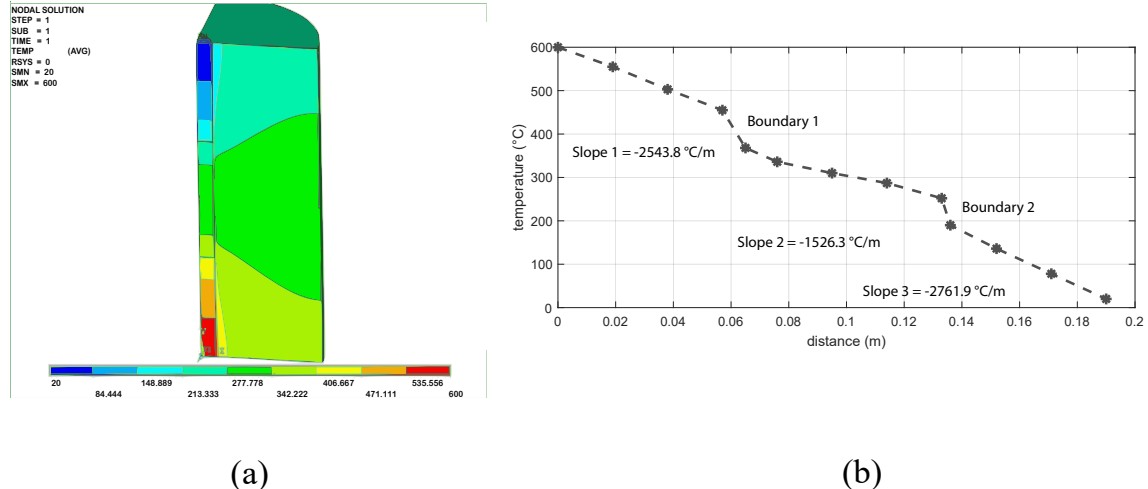

(a)                                                                                          (b)

**Figure 12.** Results for 3D analysis at a test temperature of 310 °C. (**a**) Temperature distribution in °C. (**b**) Graphic of temperatures of the symmetry line.

However, the Figure 13 shows the comparison between 2D and 3D models. It was found the maximum difference is 40 °C, and the minimum is 0.6 °C in the two analyses. In this case, there is no experimental evaluation, because with the information obtained in this work, it is possible to evaluate if it is necessary to make modifications to the existing bar system to implement the new heater, due to the temperature reached in the system that includes the bars and the insulator [20,21].

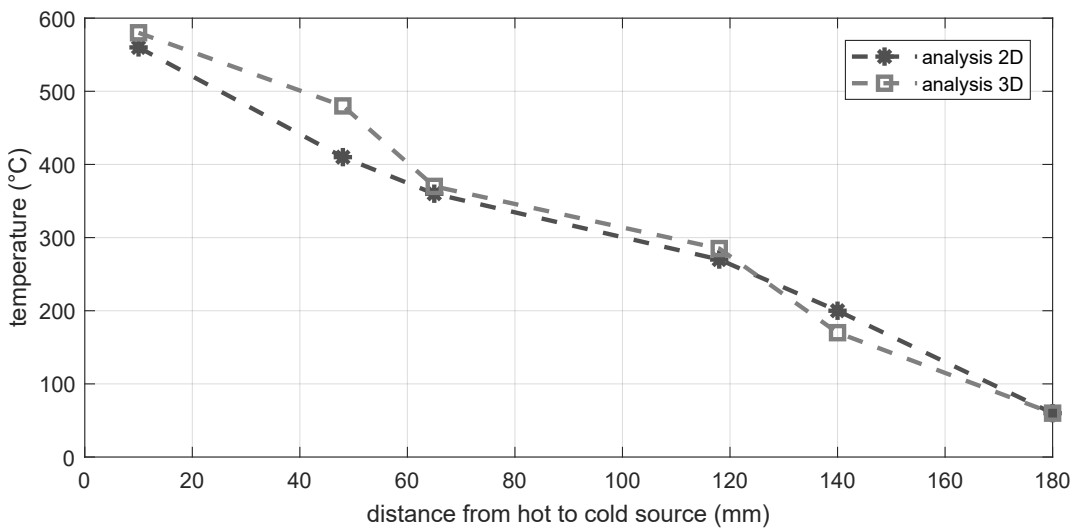

**Figure 13.** Comparison of results for 2D and 3D at a test temperature of 310 °C in thermocouple positions.

## 5. Discussion and Conclusions

### 5.1. Discussion

The vast majority of technological advances achieved in modern society have been supported by the discovery and development of engineering materials and manufacturing processes used to obtain them. An adequate selection of materials and methods guarantee the designers of mechanical

parts their correct functioning, i.e., the performance of the designed components [22]. By means of FEM, it was intended to develop a model that could verify an adequate integration of all the necessary inputs in the heat transfer analysis. An analysis was carried out where the whole model consisted of different materials, and after applying the calculated thermal limit conditions, values were found that produces a close approximation to the experimental results. Some observed discrepancies can be attributed to inaccuracy in thermocouple locations [23–25]. Most methods are based on the availability of a wide range of materials, which must be analysed and refined, either with the help of recommendations, i.e., traditional methods, material maps with graphic method or information found in bibliographic sources or in software by virtual databases, type of material, which should result in the most appropriate for the intended purpose.

In this work, the levels of correspondence of the experimental results concerning those obtained by numerical simulation are outstanding. According to work in [26–28], the uncertainty that was reached in the measurement of thermal conductivity is less than 5%, of which more than 90% of the contribution to the uncertainty corresponds to the reference material; therefore, the temperature measurement does not contribute significantly with the final value of the uncertainty of the thermal conductivity of the material under test. Therefore, the results obtained are acceptable due to their little impact on the total value of the uncertainty of thermal conductivity [29].

The results obtained with the designed equipment have been validated using a comparative analysis with the values obtained according to ASTM E1225-99 "Standard Test Method for thermal conductivity of solids by the guarded comparative longitudinal heat flow" [9]. These tests were carried out at three test temperatures:

- 1–310 °C, to achieve this, the temperature of the hot source was set at 600 °C and the temperature of the cold source at 20 °C.
- 2–175 °C, to reach this value, the temperature of the hot source was set at 339 °C and the temperature of the cold source at 20 °C.
- 2–150 °C, to achieve this, the temperature of the hot source was set at 279.5 °C and the temperature of the cold source at 20 °C.

Regarding the work in [30,31], the sources of the uncertainty values are compared with the graphs obtained in this work. It is shown that the temperature difference near the heat source and the cold source are those that present a more significant deviation concerning the experimental results. A correlation can be inferred for the contribution of uncertainty. According to [32,33], the simulations by FEM performed, where the gradients are more significant, and strictly the heat leaks in the numerical model are not being considered, which could be taken as a reference to calculate heat losses and add a correction in the final uncertainty value [34]. On the other hand, the prediction of the temperature values that the CENAM cut-off bar equipment reaches when the heater operates maximum temperature makes it possible for an adequate selection of material for the fibreglass surface, because the area with higher temperature can reach up to 386°C according to the error obtained in this work [35].

## 5.2. Conclusions

Thermal conductivity is important in several applications to different temperatures, i.e., aerospace industry; nuclear industry; nuclear control rods; radioactive waste containment; the phase change material; or items such as bearings, piston parts, pumps, compressor plate valves, cable insulation and medical implants used in different applications. Therefore, it is essential to measure this thermal property with the most accurate as possible.

The percentage error obtained by ANSYS was 13.5% for the robust model, averaging the 4 volumes (2 copper bars, 1 aluminium bar, and the fibreglass insulator).

The union of elements that interact between the interfaces of the materials is essential and considerably affects the results. In spite of this, it is possible to know with a 13.5% error the temperature distribution inside the system of cut bars [35].

The values near the borders are very far from the experiment; however, the values near the sample bar are too close to those obtained experimentally.

The temperature deviation obtained through simulation and experimental work of the cold source is affected by the contact between it and the copper bar. On the other hand, the same is valid for hot source contact. In the simulation, losses due to bad contact or heat leakage to the environment are not considered. Other methods proposed in the literature to characterise materials have allowed us to verify experimental results [36,37].

The bars cut in CENAM must be designed to prevent radial heat leakage because, according to the simulation results, there is heat leakage in this direction. With the results obtained, a guard can be proposed that balances the gradients generated in the system. Because three distinct sections are noted for the latter case, it would be 213.33 °C, 277.77 °C, and 342.22 °C. As a proposal for improving the design of the CENAM cut bar system.

According to the analysis of the results and the simulations obtained, the following design criteria are proposed. Improve the thermal contact between the hot source and the cold source with the reference bars, which can be achieved by a system that compresses the three bars. Implement a guard with a control system in the hot source and in the cold source, which, although it does not eliminate the radial temperature gradient, reduces it to a minimum. From simulation realised, it is possible to obtain the location of the temperature sensor for the control guard system.

## 6. Future Work

ASTM standard E1225-99 establishes that the measuring equipment by the cut-car method can operate at 1000 °C with a fairly acceptable uncertainty of less than 2%. Then the equipment used by CENAM needs major adjustments and, most likely, a redesign because other critical heat transfer phenomena such as radiation have to be considered. Therefore, with the support of the finite element method, it is intended to analyse the behaviour of a new system, but at an operating temperature of 1000 °C to develop a measuring device that operates at that temperature and can perform measurements of thermal conductivity at temperatures of 500 °C.

There is another problem that affects the accuracy of the results obtained for thermal conductivity value by the method presented in this work, and they are the radial heat leaks. With the use of the finite element method, we will try to minimise these heat leaks to increase accuracy. Another future work is to try to calculate the heat losses by comparing the experimental method and the simulation to obtain an estimate of the heat losses in the experimental system and to find the cause of them. It will serve to make corrections when calculating the uncertainty and implement improvements to the system to reduce these heat leaks.

**Author Contributions:** Conceptualization, J.E.E.G.D. and O.J.G.-R.; Methodology, M.A.Z.-A. and J.R.-R.; Writing–original draft preparation, N.M.-L., R.G.G. and J.R.-R.; Writing–review and editing, M.A.Z.-A., J.E.E.G.D., J.R.-R. and O.J.G.-R.; Supervision, J.R.-R. and J.E.E.G.D.; Data curation, D.J.G.M. All authors have read and agreed to the published version of the manuscript.

**Funding:** This research was partial funded by CONACYT and PRODEP.

**Acknowledgments:** The authors appreciate the support of Centro Nacional de Metrología (CENAM). The authors appreciate Master César Javier Ortiz Echeverria for his support during the revision of the project.

**Conflicts of Interest:** The authors declare no conflicts of interest.

## Nomenclature

| | |
|---|---|
| $\lambda_{C_u}$ | Thermal conductivity of the copper material |
| $\lambda_M$ | Thermal conductivity of the sample |
| $\lambda_{R_1}$ | Thermal conductivity of the reference material 1 |
| $\lambda_{R_2}$ | Thermal conductivity of the reference material 2 |
| $\lambda_z$ | Thermal conductivity of any material |
| $\Delta_{T_i}$ | Temperature gradient ΔT through an area A (the area through which heat flows) |
| $\Delta_{T_1}$ | Temperature difference among reference material 1 |
| $\Delta_{T_3}$ | Temperature difference among reference material 2 |
| $\Delta_{T_2}$ | Temperature difference among sample bar |
| °C | Celsius degrees |
| Al | Aluminium |
| ASTM | American Society for Testing and Materials |
| CENAM | Centro Nacional de Metrologia |
| CST | Cold Source Temperature |
| Cu | Copper |
| emf | electromotive force |
| FEM | Finite element method |
| HST | Temperature of hot source |
| m | meter |
| mm | millimetre |
| PVC | Polyvinyl chloride |
| $r_A$ | Bar radius |
| $r_B$ | Guard radius |
| TFF | Heat sink Temperature |
| $T_i$ | Temperature in each of the $z_i$ positions where thermocouples are placed |
| $x$ | Denotes approximate thermocouple positions |
| $y$ | Denotes axis $y$ in cartesian coordinate system |
| $z_i$ | Reference distance for thermocouple location $T_i$ in the system |

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
