# Peer review of "Finite Element Method and Cut Bar Method-Based Comparison Under 150°, 175° and 310 °C for an Aluminium Bar"

_applsci, doi:10.3390/app10010296_

Round 1
Reviewer 1 Report
Review:
Finite Element Method and Cut Bar Method – based Comparison Under 150 ° ,175° and 300°C for an Aluminium Bar
The aim of the paper is using of thermal conductivity measurement system for solid conductive metalic materials limited in its operating intervals to measurements of maximum 300 °C . Simulations were performed to compare the distribution of temperatures developed in the measurement system as well as the radial heat leaks, which affect the measurement parameters for an aluminum bar, and uses copper bars as reference material.The experiments on the measurement system of CENAM were realized by bar cut method and FEA anlysis was realized as the simulation method.
Comments:
Row 66, 100: Please describe the parameter of the termocuples or/and the producer. And country of the producer.
Row 68: Which sort and quantity of the force did you used.
Row 76: How the heating are realised.
Row 80: Probably will be better use term difference temperature as the average temperature.
Row 81: How the acquisition of the date is realised.
Row 82: How do you know, that the staedy-state was reached.
Row 90, Eq. 1: Please explain what does it mean ΔTi.
Row 100: Please write the type, producer and country of the producer of the multimeter and the scaner.
Row 101: Please describe the software for data acquisition in detail.
Row 108: The λz is not defined.
Row 114: Please describe the experimental data evaluation and the Eq. 3 determination.
Row 119: Please write the version and producer of ANSYS, the used modules.
Row 126:Please describe the boundary conditions and temperature loads of the solved model in ANSYS.
Row 137: Table one is not clear. Please explain all quantities and theirs role as the boudary and initial conditions.
Row 156: How did you determined the variations of the temperature. What were the deviations of the eperimental and computed quantities.
The question is partially original and well defined. Some area e. g. „computational model setup“ are not well defined. The thermal transient analysis is not described teoretically. The results provide an advance in current knowledge.
The results are interpreted appropriately. They are significant. All conclusions are justified and supported by the results
Introduction and Technique bacground are desribed only by seven presentations. The data and analyses presented partially appropriately The highest standards for presentation of the results was used.
Study is correctly designed.The analyses are performed with the highest technical standards – FEM analysis. The data are robust enough to draw the conclusions. Experimental determination of boundary conditions of Eq. 3 are not described. The uncerntainty of the experimental method and model is not realised sufficiently. Authors describe the error of ANSYS model only in Conclusions.
The conclusions are interesting for the readership of the Journal. The paper could be interesting only to a limited number of people interesting with the heat transfer or FEM anlysis or material engineering.
The measurements are standard, but very usuful. The FEM analysis is also standard methods. The obtained result are new knowledge.
English language is appropriate and understandable.
Author Response
Row 66, 100: Please describe the parameter of the termocuples or/and the producer. And the country of the producer.
Thank you very much for the comment. The change is described in lines 67-100.
Row 68: Which sort and quantity of the force did you used.
Thanks for the observation. The adjustment was made on lines 67 to 68.
Row 76: How the heating is realized.
Thanks for the observation. The adjustment was made on lines 78 to 80.
Row 80: Probably will be better using the term difference temperature as the average temperature.
Thank you very much for the comment. The changes were made, see lines 82 to 86.
Row 81: How the acquisition of the data is realised.
Thank you for the comment. The changes were made, see lines 80 to 86
Row 82: How do you know that the staedy-state was reached.
Thank you very much for your observation. Adjustments were made on lines 90 to101.
Row 90, Eq. 1: Please explain what does it mean ΔTi.
Thank you very much for the observation. The meaning is included in the nomenclature.
Row 100: Please write the type, producer and country of the producer of the multimeter and the scanner.
Thank you very much for the comments. It is explained in lines 113 to 123.
Row 101: Please describe the software for data acquisition in detail.
Thank you very much for the comments. It is explained in lines 113 to 123.
Row 108: The λz is not defined.
Thank you very much for your observation. The meaning is included in the nomenclature.
Row 114: Please describe the experimental data evaluation and the Eq. 3 determination.
Thank you very much for the comments. It is explained in lines 139 to 140.
Row 119: Please write the version and producer of ANSYS, the used modules.
Thank you very much for the comments. It is explained in lines 142 to 143.
Row 126:Please describe the boundary conditions and temperature loads of the solved model in ANSYS.
Thank you very much for the comments. It is explained in lines 142 to 147.
Row 137: Table one is not clear. Please explain all quantities and their role as the boudary and initial conditions.
Thank you very much for the observation. The explanation is included in lines 163 to 166.
Row 156: How did you determine the variations of the temperature. What were the deviations of the experimental and computed quantities.
Reviewer 2 Report
This paper is well-written and can be accepted
Author Response
This paper is well-written and can be acceptedThank you very much for the comment.
Reviewer 3 Report
The authors used experiments and finite element analysis to study the thermal conduction in solid materials. The 2D and 3D simulations were carried out to calculate the distribution of temperatures in the aluminum bar at the temperature of 150◦ , 175◦ and 310◦C. The results were further compared with the experimental measurements and showed fair agreement.
The reviewer suggests the paper be published after carefully addressing the following issues.
1. Please briefly derive equation 2 and explain
2. The authors measure the temperature at z2 and z3, z4 and z5, why not measure the temperature at the junction of each material?
3. Is there any reference for Equation 3?
4. In Figures 5, 6, 8, 9, 11, 12, please mark the boundary of each material and indicate the value of slope on the curve.
5. Figure 6, in the caption, shouldn’t be the result of 3D analysis?
6. Can the authors explain more clearly why the 3D model has a distinct transition between each material in the temperature-distance curve?
7. Please check the typo through the manuscript. For example, "The 3 bars are mounted vertically, the two reference ones ate the ends and the test one in the middle". "Ate" should be "at".
Author Response
Please briefly derive equation 2 and explainThank you very much for the comment lines 96 to 99.
The authors measure the temperature at z2 and z3, z4 and z5, why not measure the temperature at the junction of each material?Thank you very much for your observation. The explanation was included in lines 132 and 133.
Is there any reference for Equation 3?Thank you very much for your observation. The reference to Equation 3 was included inline 139.
In Figures 5, 6, 8, 9, 11, 12, please mark the boundary of each material and indicate the value of slope on the curve.Thank you very much for the comments.
Figure 6, in the caption, shouldn’t be the result of 3D analysis?Thank you very much for your observation. The explanation was made in lines 181 to 183.
Can the authors explain more clearly why the 3D model has a distinct transition between each material in the temperature-distance curve?Thanks for the observation. The adjustment was made in lines 182 to 184.
Please check the typo through the manuscript. For example, "The 3 bars are mounted vertically, the two reference ones ate the ends and the test one in the middle". "Ate" should be "at".Thanks for the observation. Typo mistakes have been corrected.